# A Novel Approach For Adversarial Robustness

## Abstract

Deep learning has made tremendous progress in the last decades; however, it is not robust to adversarial attacks. To deal with this issue, perhaps the most effective approach is adversarial training at a high computational cost, although it is impractical as it needs prior knowledge about the attackers. In this paper, we propose a novel approach that can train a robust network only through standard training with clean images without awareness of the attacker's strategy. Essentially, we add a specially designed network input layer, which accomplishes a randomized feature squeezing to greatly reduce the malicious perturbation. It achieves the state of the art of robustness against unseen $l_1, l_2,$ and $l_\infty$-attacks at one time in terms of the computational cost of the attacker versus the defender through just 100/50 epochs of standard training with clean images in CIFAR-10/ImageNet.

## 1 Introduction

The vulnerability of neural networks has been widely acknowledged by the deep learning community since the seminal work of Szegedy et al. (2014). A lot of solutions have been proposed to solve these problems. They can be categorized into three classes.

The first is preprocessing-based approaches which include bit-depth reduction (Xu et al., 2018), JPEG compression, total variance minimization, image quilting (Guo et al., 2018), and Defense-GAN (Samangouei et al., 2018). With this kind of preprocessing, the hope is that the adversarial effect can be reduced. However, it neglects the fact that the adversary can still take this operation into account and craft an effective attack through Backward Pass Differentiable Approximation (BPDA) (Athalye et al., 2018).

Secondly, perhaps the most effective method is adversarial training. The idea is straightforward. In the training phase, the attack is mimicked through the backward gradient propagation with respect to the current network state. There is a large volume of work that falls into this class which differs in how to generate extra training samples. Madry et al. (2018) used a classical 7-step PGD attack, while other approaches are also possible, such as Mixup inference (Pang et al., 2020), feature scattering (Zhang & Wang, 2019), feature denoising (Xie et al., 2019), geometry-aware instance reweighting (Zhang et al., 2021), and channel-wise activation suppressing (Bai et al., 2021). External (Gowal et al., 2020) or generated data (Gowal et al., 2021; Rebuffi et al., 2021) are also beneficial for robustness, and based on which parameterizing activation functions (Dai et al., 2022) can do further improvement. Theoretically principled trade-off between robustness and accuracy is analyzed in Zhang et al. (2019), which is somehow reconciled by a self-consistent robust error (Pang et al., 2022) or reducing excess margin along certain adversarial directions (Rade & Moosavi-Dezfooli, 2022). Pre-training is also helpful in Hendrycks et al. (2019). Recently, Jin et al. (2022) proposed to enhance adversarial training with second-order statistics of weights. The inherent drawback is the large computation cost, therefor practical significance is somehow diminished. It should be noted here that there do exist some free or fast adversarial training schemes as in Shafahi et al. (2019); Wong et al. (2020) or an improved subspace variant (Li et al., 2022), but there is some degradation in performance. Another big issue is that the adversarial training needs to know some prior knowledge about attacks, otherwise a simulation of attack can not be conducted. This is certainly not realistic in practice. Usually, they only do training with just one particular $l_p$-attack, with the exception that Laidlaw et al. (2021) uses Perceptual Adversarial Training against multiple attacks. Also, there is a possibility of robust overfitting (Rice et al., 2020).

The last is adaptive test-time defenses. They try to purify the input in an iterative way as in Mao et al. (2021); Shi et al. (2021); Yoon et al. (2021) or adapt the model parameters or even network structures to reverse the attack effect. For example, close-loop control is adopted in Chen et al. (2021), and a neural Ordinary Differential Equation (ODE) layer is applied in Kang et al. (2021). Unfortunately, most of them are proven to be not effective in Croce et al. (2022).

It turns out the progress is not optimistic, and even 1%-2% improvement on AutoAttack(Croce & Hein, 2020) requires huge computational cost and moreover, not effective for unseen attacks. Here we ask a question: "can we design a novel network and loss function thereof that can drive the network to be robust on its own without awareness of adversarial attacks?" In other words, we do not intend to generate extra adversarial samples like most other approaches do, and standard training with clean images is enough. Indeed, there should be no prior knowledge of attacks needed at all.

This certainly poses a great challenge to the construction of networks as it is not clear even whether it is feasible. On the other hand, it appears to be possible since deep networks have a very high capacity. Unfortunately, Ilyas et al. (2019) pointed out network tends to learn discriminant features that can help correct classification, regardless of robustness. It motivates us to take the point of view from the network input side. How can we make a new input layer that is most suitable for network robustness? Our intuition is essentially very simple. As attacks can always walk across the class decision boundary through the malicious feature perturbations, it appears that feature squeezing might be helpful, at least reducing the space of being altered. However, fundamentally different from the work (Xu et al., 2018), we squeeze the input features in a random and controlled way with parameters learned during training as shown in Figure 1, which will be elaborated in the latter sections. The experiments of CIFAR-10 and ImageNet demonstrate this approach is very useful in promoting robustness of networks.

In summary, we present an efficient approach that achieves the state of the art of robust accuracy when attack computations are constrained, especially for black-box and $l_1, l_2$-attacks, only through standard training with clean images, without any prior knowledge about the attacks.

## 2 RELATED WORKS

There are some works that add some extra preprocessing steps. For example, in Yang et al. (2019), pixels are randomly dropped and then reconstructed using matrix estimation. Ours is not preprocessing. We just add an extra layer inside the network, and the network is trained and tested as usual without explicit image completion. Besides this, to get high robust accuracy, Yang et al. (2019) needs adversarial training while we adopt standard training with clean images.

Another related work is certified adversarial robustness via randomized smoothing (Cohen et al., 2019). The base classifier is trained with Gaussian data augmentation, and inference is based on the most likely class of the input perturbed by isotropic Gaussian noise. Ours is based on standard training and test, and there is no perturbation-based training data augmentation involved at all.

Recently, there are some works that address the robustness from the network architecture's perspective. Wu et al. (2021) investigates impact of the network width on the model robustness, and proposes Width Adjusted Regularization. Similarly, Huang et al. (2021) explores architectural ingredients of adversarially robust deep neural networks in a thorough manner. Liu et al. (2023a) established that the higher weight sparsity is beneficial for adversarially robust generalization via Rademacher complexity. Wang et al. (2022) proposes batch normalization removal, such that adversarial training can be improved. Singla et al. (2021) shows that using activation functions with low curvature values reduces both the standard and robust generalization gaps in adversarial training. It is in some sense similar to ours, but our motivations are fundamentally different. There is no adversarial training involved in our approach at all.

Robust Vision Transformer has been advocated in Mao et al. (2022). Under the setting of standard training, it is better than previous Vision Transformers and CNNs, however, unfortunately, not comparable with the adversarial training methods, which are surpassed by ours.

Regularization has also been widely adopted in adversarial training. Cui et al. (2021) uses model logits from one clean model to guide learning of another robust model. Spectral norm regularization based on Lyapunov theory has also been proposed in Rahnama et al. (2020) to improve the robust-

ness against $l_2$ adversarial attack. Compared with these regularization methods, ours seeks for better network input layer design to make the network become robust on its own.

## 3 BACKGROUND

A standard classification can be described as follows:

$$\min_{\vartheta} E_{(x,y)\sim D} \left[ L\left(x, y, \vartheta\right)\right], \tag{1}$$

where data examples $x \in R^d$ and corresponding labels $y \in [k]$ are taken from the underlying distribution $D$, and $\vartheta \in R^p$ is the model parameters to be optimized with respect to an appropriate function $L$, for instance cross-entropy loss. When $x \in R^d$ can be maliciously manipulated within a set of allowed perturbations $S \subseteq R^d$, which is usually chosen as a $l_p$-ball ($p \in \{1, 2, \infty\}$) of radius $\epsilon$ around $x$, Equation 1 should be modified as:

$$\min_{\vartheta} E_{(x,y)\sim D} \left[ \max_{\delta \in S} L\left(x + \delta, y, \vartheta\right)\right]. \tag{2}$$

An adversary implements the inner maximization via various white-box or black-box attack algorithms, for example, APGD$_{\text{ce}}$ (Croce & Hein, 2020) or Square Attack (Andriushchenko et al., 2020). The basic multi-step projected gradient descent (PGD) is

$$x^{t+1} = \Pi_{x+S}\left(x^t + \alpha \text{sgn}\left(\nabla_x L\left(x, y, \vartheta\right)\right)\right), \tag{3}$$

where $\alpha$ denotes a step size and $\Pi$ is a projection operator. In essence, it uses the current gradient to update $x^t$, such that a better adversarial sample $x^{t+1}$ can be obtained. Some heuristics can be used to get better gradient estimation in Croce & Hein (2020). On the other hand, outer minimization is the goal of a defender.

Adversarial training is the most effective approach to achieve this outer minimization via augmenting the training data with crafted samples. In fact, all current approaches, including test-time adaptive defense as it needs a base classifier, aim to learn the parameters of a pre-existing model to improve the robustness. In this paper, we try to increase the robustness through a specially designed input layer such that standard training with clean images can be adopted.

## 4 METHOD

### 4.1 INPUT LAYER

As we stated earlier, the goal of input layer is to squeeze the input feature in a random and controlled way. The whole procedure is depicted in Figure 1.

It consists of the following steps:

1. The input $x$ with $r, g, b$ channel will be normalized to a variable with a mean 0 and a standard deviation 1, through $\tilde{x} = \frac{x - mean}{std}$ in the input layer.

2. The normalized value $\tilde{x}$ goes through a $3 \times 3$ 2D convolution and ReLU, and we get $\hat{x}$ with three channels.

3. The final output $y$ is a Sigmoid of the element-wise three-term multiplication, $(\tilde{x} + \varepsilon) \times (\hat{x} - \delta) \times \left(\frac{1}{\hat{x}+\gamma}\right)$. Here $\varepsilon$ is a Gaussian random variable with a mean 0 and a standard deviation $\sigma$; $\delta$ is a uniform one on $[0, 1]$; and $\gamma$ is a small constant in order to make the denominator always positive, which is $1 \times 10^{-5}$ in this paper.

So essentially,

$$y = \frac{1}{1 + \exp\left(-\frac{(\tilde{x}+\varepsilon) \times (\hat{x}-\delta)}{\hat{x}+\gamma}\right)}. \tag{4}$$

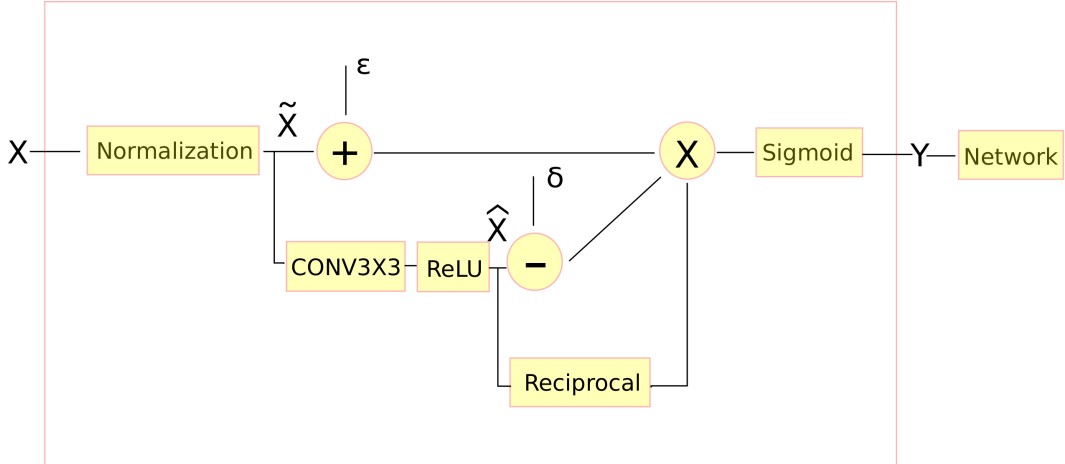

Input Layer

Figure 1: Our specially designed input layer is inside the red rectangle. The input image $x$ is first normalized, then undergoes three paths. On one path, Gaussian noise $\epsilon$ is added, and the other two paths include $3 \times 3$ convolution and ReLU followed by subtraction of noise $\delta$ which has the uniform distribution on $[0, 1]$, and reciprocal respectively. Finally, all three terms are combined through multiplication and the result feeds to the Sigmoid. The final Y will be used as inputs to the classification network, the same as other training approaches. End-to-end training scheme is adopted to learn the parameters of $3 \times 3$ convolution.

This formula can be interpreted this way. $\tilde{x} + \varepsilon$ is a polluted version of the input image,and $\frac{\hat{x} - \delta}{\hat{x} + \gamma}$ tries to modulate the image based on the $\hat{x}$, named as sampling matrix having the same size as input $x$. Due to the ReLU operation, $\hat{x}$ is always non-negative. Since $\delta$ has uniform distribution on $[0, 1]$, the numerator $(\tilde{x} + \varepsilon) \times (\hat{x} - \delta)$ can be considered as a random fraction $\hat{x} - \delta$ of $\tilde{x} + \varepsilon$, which will be allowed to feed to Sigmoid.

The key motivation is that if we enforce $\hat{x}$ to be very small through some loss function, the whole $\frac{(\tilde{x} + \varepsilon) \times (\hat{x} - \delta)}{\hat{x} + \gamma}$ will become big and the response of Sigmoid will be on the saturated region, i.e., most elements of $y$ will be either 0 or 1. In other words, the input feature will be squeezed in a random manner where the parameters of sampling matrix $\hat{x}$ are learned on the end-to-end training.

## 4.2 LOSS FUNCTION

As mentioned earlier, we have to design a loss function to implement our motivation to make the sampling matrix $\hat{x}$ small. For each $\hat{x}$ , we get $S$, the average of all the elements of $\hat{x}$ that are greater than some threshold $\beta$. A small $\beta$ means $\hat{x}$ will become sparse. The final loss function is:

$$L = \alpha \times L_{ce} + S, \tag{5}$$

where $L_{ce}$ is cross-entropy loss, and $\alpha$ is the weight. When $\alpha$ becomes large, the loss function falls back to standard cross-entropy.

In summary, there are only three parameters, $\sigma$ of Guassian noise, threshold $\beta$, and weight $\alpha$.

## 4.3 LAST MOVE

We emphasize here that since our approach is random, a same sample could be classified with different logits when executed multiple times. It will seriously mislead attackers, which will report a wrong robust accuracy. For that reason, we always take the last-move advantage. In other words, in test time, we always take the adversarial samples generated by attackers and feed them to our network once again to test. We think that is fair in practice. Attackers can always take an arbitrarily long time to figure out a malicious sample, but they have only one chance to submit it. It is the

| Paper | Clean | AA-$l_\infty$ | AA-$l_1$ | AA-$l_2$ | Square-$l_\infty$ | Square-$l_1, l_2$ |
|---|---|---|---|---|---|---|
| Wang et al. (2023)[#] | 92.44 | 67.31 25.46 | 10.23 | 1.18 | 73.57 40.28 | 35.77 30.78 |
| Gowal et al. (2021)[#] | 87.50 | 63.38 27.91 | 10.85 | 1.94 | 68.90 40.91 | 35.71 30.15 |
| Dai et al. (2022)[#] | 87.02 | 61.55 26.28 | 11.22 | 1.98 | 66.99 38.86 | 37.15 30.26 |
| Wang et al. (2023)* | 95.16 | 49.33 3.86 | 46.08 | 6.59 | 67.02 18.69 | 69.38 44.20 |
| Rebuffi et al. (2021)* | 91.79 | 47.83 5.04 | 42.80 | 8.23 | 62.45 19.73 | 65.66 42.66 |
| Laidlaw et al. (2021) | 82.40 | 30.20 4.50 | 32.40 | 7.10 | 46.40 15.30 | 53.30 34.20 |
| Ours | 81.88 | **80.43 77.01** | **78.34** | **63.10** | **80.87 78.31** | **81.59 80.58** |

Table 1: AutoAttack comparison on CIFAR-10 (WideResNet-28-10 only except ResNet-50 in Laidlaw et al. (2021)). * denote models that are trained with $l_2$-$\epsilon$=0.5, while [#] with $l_\infty$-$\epsilon$=8/255; both * and [#] need extra training data. $l_\infty$-$\epsilon$=8/255, 16/255; $l_1$-$\epsilon$=12 and $l_2$-$\epsilon$=2. The bold indicates the best for each column.

attacker's responsibility to provide a stable adversarial sample, and the last move is always the defender's privilege.

## 5 EXPERIMENTS

To verify the effectiveness of our approach, we conducted the experiments on CIFAR-10 and ImageNet. Both the threshold $\beta$ and the weight $\alpha$ are set to be 0.1 uniformly in our study, while $\sigma$ of Guassian noise is different for two datasets which will be addressed in the following, as this parameter essentially is related to clean accuracy which in turn depends on dataset and network architecture. We evaluate on AutoAttack and Square Attack of $l_\infty$, $l_1$ and $l_2$. AutoAttack is comprised of four attacks, namely Auto-PGD for cross-entropy and Difference of Logits Ratio (DLR) loss, FAB-attack (Croce & Hein) and the black-box Square Attack (Andriushchenko et al., 2020), and commonly used as a robustness evaluator. Square is used as a representative black-box attack separately as well, as it is of practical significance.

### 5.1 CIFAR-10

In this paper, we choose the wide residual network WideResNet-28-10 (Zagoruyko & Komodakis, 2016) as the base network, where we add our specially designed input layer as described in Section 4. $\sigma$ of Guassian noise is 0.5. The initial learning rate of 0.1 is scheduled to drop at 30, 60, and 80 out of 100 epochs in total with a decay factor of 0.2. The weight decay factor is set to $5\times10^{-4}$, and the batch size is 200. To emphasize again, we only perform standard training through just 100 epochs.

We compare our method with some state of the arts, which are all based on adversarial training. $l_\infty$, $l_1$ and $l_2$-AutoAttack (Croce & Hein, 2020) are adopted. Some results are shown in Table 1. All these models are trained with one particular type of attack either with $l_\infty$-$\epsilon = 8/255$ or $l_2$-$\epsilon = 0.5$, except Laidlaw et al. (2021) adopts neural perceptual threat model.

Ours outperforms all other methods significantly against multiple unseen attacks including the practical black-box Square Attack, although we only use standard training with clean images. Indeed, robustness against multiple attack models should be vital for applications since we can't assume the attack will follow the simulations conducted in the malicious sample generation in adversarial training methods. Unfortunately, most current works fail to generalize well to unseen attacks.

Another significant advantage of ours is the computational cost shown in Table 2, where all other competitors in Table 2 are 3-5 orders of magnitude higher than ours.

As our algorithm is random in nature, we also adopt the EOT-test as shown in Table 3. There are some drops in accuracy, however, it is still much better than others. Note that EOT incurs a large computational cost, so actually, it is unfair to compare the robustness of networks without computation constraints.

| Paper | #Extra | #Epochs | #PGD | #Cost |
|---|---|---|---|---|
| Wang et al. (2023)[#] | 20M | 2400 | 10 | $9.6 \times 10^4$ |
| Gowal et al. (2021)[#] | 100M | 2000 | 10 | $4 \times 10^5$ |
| Dai et al. (2022)[#] | 6M | 200 | 10 | $2.4 \times 10^3$ |
| Wang et al. (2023)* | 50M | 1600 | 10 | $1.6 \times 10^5$ |
| Rebuffi et al. (2021)* | 1M | 800 | 10 | $1.6 \times 10^3$ |
| Ours | 0 | 100 | 0 | 1 |

Table 2: Computational cost comparison. Excluding the cost of gathering extra data, the training cost in #Cost is roughly the product of #Epochs(training epochs), #Extra, and #PGD(pgd steps adopted in adversarial inputs generation) with respect to ours, i.e., 50K inputs and 100 epochs of standard training, which is denoted by 1.

| Attacks | $APGD_{ce}$ | $APGD_{dlr}$ |
|---|---|---|
| $l_\infty$-$\epsilon$=8/255 | 75.96 | 77.46 |
| $l_\infty$-$\epsilon$=16/255 | 57.92 | 64.48 |
| $l_1$-$\epsilon$=12 | 67.89 | 67.68 |
| $l_2$-$\epsilon$=2 | 47.18 | 55.74 |

Table 3: The EOT accuracy of $APGD_{ce}$ and $APGD_{dlr}$ attacks for CIFAR-10.

## 5.2 IMAGENET

ImageNet is the most challenging dataset for adversarial defense. In this paper, ImageNet only refers to ImageNet-1k without explicit clarification, and robustness is only evaluated on the 5000 images of the ImageNet validation set as in RobustBench (Croce et al., 2021). For simplicity, we choose the architecture of ConvNeXt-T + ConvStem in Singh et al. (2023). Our training scheme is very simple. All parameters are randomly initialized, followed by standard training for 50 epochs with heavy augmentations without CutMix (Yun et al., 2019) and MixUp (Zhang et al., 2018), as these will undermine the viability of our sampling matrix. While for the same ConvNeXt-T + ConvStem in Singh et al. (2023), although ConvStem is randomly initialized, the ConvNeXt-T part is from a strong pre-trained model which usually takes about 300 epochs. Thus the whole network needs extra standard training for 100 epochs to get good clean accuracy, followed by 300 epochs of adversarial training with 2-step APGD. So the total cost is up to $300 + 100 + 300 \times (2 \, (for \, APGD \, steps) + 1 \, (for \, weights \, update)) = 1300$ , which is around $1300/50 = 26$ times bigger than ours.

| Architecture | Clean | AA-$l_\infty$ | | AA-$l_1$ | AA-$l_2$ | Square-$l_\infty$ | | Square-$l_1, l_2$ | |
|---|---|---|---|---|---|---|---|---|---|
| **ConvNeXt-T + ConvStem** | | | | | | | | | |
| Singh et al. (2023) | 72.74 | 49.46 | 24.10 | 24.50 | 48.40 | 63.42 | 52.44 | 49.40 | 68.06 |
| Ours | 69.92 | **65.92** | **52.64** | **68.46** | **69.44** | 69.48 | **68.52** | **69.40** | 69.28 |
| **Swin-L** | | | | | | | | | |
| Liu et al. (2023b)* | 78.92 | 59.56 | 32.72 | 26.88 | 52.02 | **70.38** | 61.56 | 55.52 | **74.18** |
| **ConvNeXt-L** | | | | | | | | | |
| Liu et al. (2023b)* | 78.02 | 58.48 | 32.00 | 26.18 | 52.22 | 70.12 | 61.04 | 54.40 | 72.86 |
| **ConvNeXt-L+ConvStem** | | | | | | | | | |
| Singh et al. (2023) | 77.00 | 57.70 | 31.86 | 22.38 | 47.02 | 69.66 | 59.48 | 54.18 | 72.80 |

Table 4: AutoAttack comparison on ImagetNet. $l_\infty$-$\epsilon$=4/255, 8/255; $l_1$-$\epsilon$=75 and $l_2$-$\epsilon$=2. The bold indicates the best for each column. * denote models that are pre-trained with ImageNet-21k.

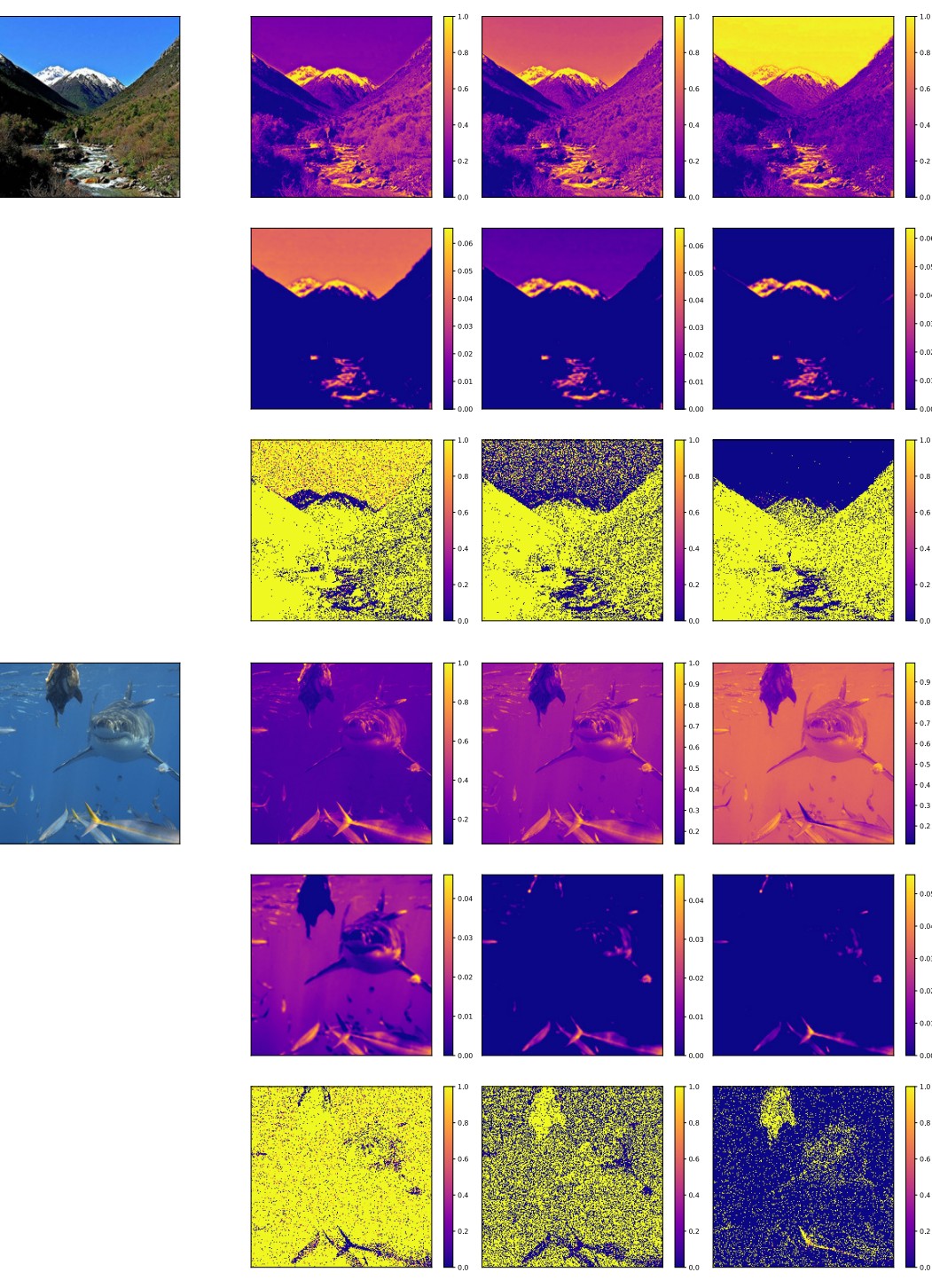

Figure 2: The two halves are arranged in a similar way. The first row shows the input valley and great-white-shark $x$, and the next two show the corresponding sampling matrix $\hat{x}$ and the final output $y$, with three channels aligned. It is very interesting to note that the continuous patterns are highly squeezed into two extreme values, 0 and 1, in $y$ due to very small $\hat{x}$. Nevertheless, the V shape pattern in valley can still be identified in $y$. Indeed, this image is classified correctly. Surprisingly, although the features of great-white-shark are buried due to our intentionally injected noise, it is classified correctly as well.

| Attacks | $APGD_{ce}$ | $APGD_{dlr}$ |
|---|---|---|
| $l_\infty$-$\epsilon$=4/255 | 45.80 | 52.50 |
| $l_\infty$-$\epsilon$=8/255 | 17.08 | 26.54 |
| $l_1$-$\epsilon$=75 | 67.58 | 68.50 |
| $l_2$-$\epsilon$=2 | 68.60 | 69.54 |

Table 5: The EOT accuracy of $APGD_{ce}$ and $APGD_{dlr}$ attacks for ImageNet.

| Architecture | Clean | Square-$l_\infty$ | Square-$l_1, l_2$ | E-Square-$l_\infty$ | E-Square-$l_1, l_2$ |
|---|---|---|---|---|---|
| **ConvNeXt-T + ConvStem** | | | | | |
| Singh et al. (2023) | 72.20 | 65.80 55.20 | 51.20 69.00 | 61.60 43.40 | 42.00 66.80 |
| Ours | 71.40 | 70.00 **70.00** | **72.20** 71.80 | **69.60 70.00** | **70.00** 70.00 |
| **Swin-L** | | | | | |
| Liu et al. (2023b)* | 79.80 | **70.60** 60.80 | 55.00 **74.40** | 66.40 52.00 | 46.80 **71.80** |

Table 6: Square Attack comparison on 500 images on validation set of ImagetNet instead of 5000 on Table 4. $l_\infty$-$\epsilon$=4/255, 8/255; $l_1$-$\epsilon$=75 and $l_2$-$\epsilon$=2. The bold indicates the best for each column. * denote models that are pre-trained with ImageNet-21k. The iterations are 5K for Square Attack, and 50K for E-Square (Enhanced-Square Attack).

As shown in Table 4, ours beats (Singh et al., 2023) by a large margin in almost all tests. To be more solid, we also compare with other methods of more sophisticated architectures, including Swin-L and ConvNeXt-L in Liu et al. (2023b), only slightly behind on Square Attack $l_\infty$-$\epsilon$=4/255 and $l_2$-$\epsilon$=2.

Some of the example feature maps in our input layers are listed in Figure 2. The input $x$, the sampling matrix $\hat{x}$, and the final output $y$ are demonstrated in three rows. Our specially designed input layer changes the input $x$ into $y$ that are extremely squeezed. On the one hand, it poses a great challenge to the network, while on the other hand, it improves the robustness.

Regarding EOT tests, the negative impacts on robust accuracy are almost negligible for $l_1$ and $l_2$, while for $l_\infty$, there is a relatively high drop. However, we stress again here that in fact, every defense is weak given sufficient computational resources. As shown in Table 6, we increase a query limit of Square Attack from 5K used in AutoAttack to 50K denoted as Enhanced-Square, and there are up to 12% decrease in robust accuracy for Singh et al. (2023), 9% for Singh et al. (2023). Interestingly, because of randomness and the last-move strategy, ours stands up, sometimes even better than clean one. Due to resource constraints, only 500 images are evaluated.

## 6 DISCUSSION

Our approach is efficient and effective; however, one may raise a big concern with respect to the obfuscated gradient or adaptive attack. Since the work of Athalye et al. (2018), the adversarial defense community is conservative about the validity of claims of effective defenses. However, Athalye et al. (2018) only investigates the attack strategies for the type of seen attacks without taking into account the computational load of the attacker, which is not sufficient. For example, adversarial training with $l_\infty = 8/255$ is usually evaluated with $l_\infty = 8/255$. In practice, the attack should not be confined to only launching the attack of the type that the defender has seen before, and the computation resources should be restricted to, for example, a certain number of queries; otherwise, the defender can reject the attack that takes too much time through other security measures. In fact, the unseen attacks are much easier to break the defense than hand-crafted and sophisticated adaptive attacks in Athalye et al. (2018), so they should come first. According to our thorough experiments, ours achieves the state of the art in this regard. Almost all previous approaches generalize poorly to unseen attacks.

The other big question may be why this approach can be so robust. The key motivation is that we try to unleash the great potential of deep networks unusually. The input features are squeezed randomly,

so the networks have to identify some robust features to get high clean accuracy; and impacts made by attacks can be minimized.

There are some limitations of this approach. Firstly, clean accuracy is lower than the state of the art. Secondly, as the sampling matrix $\hat{x}$ relies on the $3\times3$ convolution of input, it might be misled by recently proposed occlusion attack (Duan et al., 2023). Thirdly, ours is only verified by experiments, and there is no theoretical robustness guarantee.

## 7  SUMMARY

In this paper, we present a simple approach that only uses standard training with clean images, and achieves the state of the art robust accuracy on unseen $l_1, l_2,$ and $l_\infty$-attacks at one time. This method is verified through CIFAR-10 and ImageNet dataset. In the future work, we will improve the clean accuracy and take care of the occlusion attack. Theoretical analysis is also needed to better understand why it works so well.

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
