# OpenReview forum: "A Novel Approach For Adversarial Robustness"
_ICLR.cc/2024/Conference — Submitted to ICLR 2024_

### Official Review · Reviewer_88kq · 2023-10-13

**Soundness:** 1 poor
**Presentation:** 2 fair
**Contribution:** 2 fair
**Rating:** 3
**Confidence:** 4

**Summary:**

In this paper, the authors propose a new plug-in module, which can help the model against adversarial attacks. After using such a model, only standard training can fulfill the robust requirement.

**Strengths:**

1. The authors propose a random module to defend against various adversarial attacks.

2. The proposed module is lightweight without introducing too many parameters.

3. The authors consider different model structures cooperating with the proposed method.

4. There are EOT experiment results, which is important for a randomization-based defense.

**Weaknesses:**

1. Stochastic Neural Networks (SNNs) have been proposed for many years, which are not a novel approach to defend against adversarial attacks. Therefore, I think the authors overclaim their contribution.

2. The authors do not mention any related works under the topic of SNNs or other stochastic methods. I am not sure whether the authors are on purpose or not. But the Related Works should mainly discuss the most related papers.

3. The authors do not compare any SNN baselines, which causes unfairness in the experiments. For example, I can simply find a paper [1] from Google Scholar, which discusses SNN in adversarial defense. In experiments, they compared various stochastic methods, which I cannot find in this paper. This unfair comparison causes a false contribution.


[1] Yang, H., Wang, M., Yu, Z., & Zhou, Y. (2022). Rethinking feature uncertainty in stochastic neural networks for adversarial robustness. arXiv preprint arXiv:2201.00148.

**Questions:**

Please see weaknesses.

---

> ### Author Response · Authors · 2023-11-15
>
> Thank you for pointing out some SNN works. Actually, we only look for and compare with the top-ranking methods in the robustbench leaderboard https://robustbench.github.io/, where there are no SNNs. It appears that SNN is far behind the mainstream performance of adversarial defense. Of course, we will add more related work of SNN.
>
> Our approaches are very different from all SNN works, as we essentially do the random feature squeezing.
>
> Just for a quick reference, here we make some comparisons with [1,2], where, unfortunately, there is no AA evaluation and moreover no experiment with ImageNet.
>
> So we only compare with square for cifar10, In table 5 of [1], the square at 16/255 is 55.2 and 48.8 in table 4 of [2], far below ours  78.31.
>
> [1] Yang, H., Wang, M., Yu, Z., & Zhou, Y. (2022). Rethinking feature uncertainty in stochastic neural networks for adversarial robustness. arXiv preprint arXiv:2201.00148.
>
> [2] Weight-covariance alignment for adversarially robust neural networks Panagiotis Eustratiadis and Henry Gouk and Da Li and Timothy M. Hospedales ICML 2021

---

### Official Review · Reviewer_S7uL · 2023-10-18

**Soundness:** 1 poor
**Presentation:** 1 poor
**Contribution:** 1 poor
**Rating:** 1
**Confidence:** 4

**Summary:**

This paper addresses the challenge of adversarial attacks on deep learning models. The authors propose an approach that enhances network robustness without knowledge of the attack strategy. They introduce an input layer that reduces the impact of malicious perturbations, achieving robustness against various attack types through standard training with clean images on datasets like CIFAR-10 and ImageNet.

**Strengths:**

- The authors focus on an important topic, namely adversarial training to robustify deep neural networks.
- The idea of an approach that requires no prior knowledge of attacks is interesting.
- The approach is straightforward.
- The authors evaluate their approach on ImageNet.

**Weaknesses:**

- Trivial technical contribution: The manipulation of early or other intermediate features has been proposed various times in the literature. Most of these defenses have also been defeated.
- The authors do not provide any ablation studies to justify their design choices.
- The title is chosen too general and is vague.
- The writing is poor:
	- Some parts of the paper are incomprehensible.
	- The introduction is written like a related work section
	- The contributions are not clearly explained and distinguished to previous works
	- A proper explanation of the proposed approach
- I can't grasp what Figure 2 is supposed to show.
- While the authors evaluated with AutoAttack I am wondering how this method performs against the PGD-attack.
- The authors might encounter obfuscated gradients here. Hence the authors should follow the guidelines in [1] and [2] and evaluate with BPDA.
- Did the authors try if their approach also works for vision transformer models?

[1] Obfuscated Gradients Give a False Sense of Security: Circumventing Defenses to Adversarial Examples; ICML 2018
[2] On Evaluating Adversarial Robustness; ArXiv 2019

**Questions:**

Please address the points in my weakness section.

---

> ### Author Response · Authors · 2023-11-15
>
> Trivial technical contribution
>
> We don’t agree. Ours is based on random feature squeezing, which is novel and different from all other approaches.
>
> ablation studies:
>
> Essentially, our specially designed input layer is coupled with loss functions. In other words, they should be treated as a whole. So ablation studies are not applicable here.
>
> The title is chosen too general and is vague.
>
> True. At this moment, we are considering changing it to Adversarial Robustness through Random Feature Squeezing
>
> The writing is poor:
>
> We will correct it. Thanks.
>
> Figure 2:
>
> We are trying to show how each channel of input images is processed in our specially designed input layer.
>
> While the authors evaluated with AutoAttack I am wondering how this method performs against the PGD-attack.
>
> AutoAttack includes the APGD-ce and APGD -dlr, which are presumably more powerful than PGD.
>
> The authors might encounter obfuscated gradients here. Hence the authors should follow the guidelines in [1] and [2] and evaluate with BPDA.
>
> Every unit in our specially designed input layer is differentiable, so BPDA is not applicable. We do evaluate EOT.
>
> Did the authors try if their approach also works for vision transformer models?
>
> We have not tried.

---

### Official Review · Reviewer_CbH6 · 2023-11-01

**Soundness:** 1 poor
**Presentation:** 2 fair
**Contribution:** 1 poor
**Rating:** 3
**Confidence:** 4

**Summary:**

This paper proposes a input preprocessing method for defending against adversarial examples. The method works by _squeezing_ the input into the range $[0, 1]$ by first performing a simple (linear + thresholding) transformation followed by a random shift and a random scaling. Finally a sigmoid activation is used to squeeze the input to $[0, 1]$. With this simple preprocessing function, this paper reports stunning performances against strong adversarial attacks.

**Strengths:**

This paper proposes a preprocessor that obtains very strong performance on adversarial examples while not training on adversarial examples at all. This is a very surprising result if it is able to withstand thorough empirical evaluation.

**Weaknesses:**

1.	Gradient Masking and Suppression: The paper has many well-known signs of gradient masking, which is a phenomenon where estimating gradients of a classifier might be error-prone. In this paper, there are two components that might be causing this: (1) the random shift and scaling, (2) sigmoid squeezing. Further, Table 1 shows that the performance of the proposed method barely decreases under the strong auto attack ($\ell_infty, \ell_1$) as well as square attacks. This typically indicates some issues with the underlying evaluation [1].

2.	Evaluation on Black Box Attacks: Interestingly, the paper does provide an evaluation on the black box Square attack, which is considered to be a strong black box attack. However, the evaluation is subject to concern, as the performance barely dips below the benign performance after attack. A simple test to check any problems with the evaluation would be to intentionally inject adversarial examples and check if the performance is still retained [2].

3.	Code for the method and evaluations: Since the authors report stunning performance increases, in light of the above concerns, it would be easier to believe the claims if well documented code would be provided for each of the evaluations.

4.	Writing: The writing is loose and informal in some parts of the paper.
	1.	P3: Step 1: What is mean, std?
	2.	“$\delta$ is a uniform one” -> “$\delta \sim {\rm Unif}([0, 1])$", etc.
	3.	Sec 4.3: It seems that the adversarial examples are generated for one realization of the network, and tested on another. This is not standard practice.
	4.	P5: EOT — it would be useful to mention the exact parameters over EOT is performed, and how those parameters were chosen.
	5.	Figure 2: Please show the RGB in the first column, all rows — it is hard to understand what is going on by looking at R,G,B channels separately. Even then, what are we supposed to take away from this figure?

[1]: On Adaptive Attacks to Adversarial Example Defenses, Florian Tramer, Nicholas Carlini, Wieland Brendel, Aleksander Madry

[2]: Increasing confidence in adversarial robustness evaluations. Roland S Zimmermann, Wieland Brendel, Florian Tramer, Nicholas Carlini.

**Questions:**

In addition to the concerns raised above,

1.	What is the robustness vs accuracy curve? When does it dip below the benign performance, for each of the attacks tested? At what perturbation does it go to zero? At this perturbation, how does a human perform?

2.	What is the role of each of the components of the preprocessor towards the final robustness, in that what happens when each of them are replaced by an identity transformation? (1) Sigmoid, (2) Random scaling, (3) Random Shift

---

> ### Author Response · Authors · 2023-11-15
>
> P3: Step 1: What is mean, std?
>
> Mean and std are the mean and standard deviation of the training set, the parameters needed for input normalization. We just want to make everything clear, so delve into such detail which is usually ignored by many other papers.
>
> “δ is a uniform one” -> “δ∼Unif([0,1])", etc
>
> Thanks!
>
> Sec 4.3: It seems that the adversarial examples are generated for one realization of the network, and tested on another. This is not standard practice.
>
> As our approach is random, the same sample could be classified with different logits when executed multiple times. Actually, for any input clean sample, a very simple approach to defeat ours is just feeding this perhaps thousands of times, and it is highly possible one of them will fool our net. Of course, one cannot conclude that ours is useless at all. So there is nothing wrong with the last move. In fact,  other works, for example[1], also adopt this under different contexts.
>
> [1] Fighting Gradients with Gradients: Dynamic Defenses against Adversarial Attacks Dequan Wang and An Ju and Evan Shelhamer and David A. Wagner and Trevor Darrell
>
> P5: EOT — it would be useful to mention the exact parameters over EOT is performed, and how those parameters were chosen.
>
> EOT is provided by the AutoAttack toolbox, with default iterations of 20.
>
> Figure 2: Please show the RGB in the first column, all rows — it is hard to understand what is going on by looking at R,G,B channels separately. Even then, what are we supposed to take away from this figure?
>
> We want to show how each channel of input images is processed in our specially designed input layer. The first row is the input R, G, and B, the second row shows the sampling matrix with very small values due to the loss function, and the third is the output.
>
> Yes, we have tried to show the RGB in the first column, all rows when preparing the paper. But it is not very meaningful, since actually they are not the usual RGB channels of color images. But we will take into further consideration this.
>
> What is the robustness vs accuracy curve? When does it dip below the benign performance, for each of the attacks tested? At what perturbation does it go to zero? At this perturbation, how does a human perform?
>
> This is an interesting question. We will take it as our future work. In fact, EOT appears to be
> the only effective attack although at a high computation cost.
>
> What is the role of each of the components of the preprocessor towards the final robustness, in that what happens when each of them are replaced by an identity transformation? (1) Sigmoid, (2) Random scaling, (3) Random Shift
>
> When any component is missing, the method can not work. Our specially designed input layer is also coupled with loss functions.  The sampling matrix should be small such that the response of Sigmoid will be mostly on the saturated region, namely “random feature squeezing”. In other words, this is the key to the robustness.

---

> > ### Comment · Reviewer_CbH6 · 2023-11-21
> >
> > Thanks for the response. In light of the deficiencies in evaluation and writing, I have decided to keep my score.

---

### Official Review · Reviewer_S5kz · 2023-11-01

**Soundness:** 2 fair
**Presentation:** 1 poor
**Contribution:** 1 poor
**Rating:** 1
**Confidence:** 5

**Summary:**

The paper introduces a specialized input layer to improve the adversarial robustness of deep neural networks. Each pixel in the normalized images goes through a perturbation and multiplication process, and then are fed into a Sigmoid function before proceeding with the rest of the network. Evaluations on CIFAR10 and Imagenet-1k in the white-box setting demonstrate that the resulting networks are robust to various $\ell_p$ bounded perturbations generated using AutoAttack.

**Strengths:**

The proposed input layer design introduces virtually no computation overhead while improving the adversarial robustness of the network in the while-box setting, against gradient-based attacks. Empirical evaluations are performed on CIFAR10 as well as larger dataset such as Imagenet. Visualizations presented in Figure 2 are helpful in understanding the effect of the proposed input layer.

**Weaknesses:**

Figure 2 shows that the output of proposed input layers are mostly 0's and 1's, which are towards the saturation range in the sigmoid function. This means that the robustness improvement comes mostly from obfuscated gradients [1]. In other words, the network is having trouble finding effective adversarial perturbations, rather than being truly more adversarial robust compared to the baselines.

[1] Athalye et al, Obfuscated Gradients Give a False Sense of Security: Circumventing Defenses to Adversarial Examples ICML 2018

**Questions:**

A simple test to verify the obfuscating gradient behaviour is to measure whether the perturbation, found based on the attack methods in the paper, indeed reaches the specified radius of the $\ell_p$ ball.

---

### Author Response · Authors · 2023-11-15
**Summary**

Dear AC and reviewers:

Thanks for your time and effort. Most of the concern is about the obfuscated gradients or adaptive attacks. We want to stress every defense is weak given sufficient computation resources shown in Table 6. In other words, we are considering the practical significance of the defender in terms of the computational cost of the attacker versus the defender.

On the other hand, unseen attacks can also be adaptive ones. For example, as demonstrated in Tables 1 and 4, most current approaches are defeated by such unseen attacks and thus give a false sense of security in practice, the same as the effect of obfuscated gradients. Interestingly, ours stands up.

Ours achieves the state of the art in these two aspects, and the code will be released for the benefit of the adversarial defense community.

Another big concern may be related to the attackers. In other words, attackers might be fooled
by our defense such that it has found an adversarial sample due to the randomness of our net, nerveless, in fact, this sample is not stable.

Some important reviews regarding the attack.

A simple test to verify the obfuscating gradient behaviour is to measure whether the perturbation, found based on the attack methods in the paper, indeed reaches the specified radius of the ℓp ball.

Evaluation on Black Box Attacks: Interestingly, the paper does provide an evaluation on the black box Square attack, which is considered to be a strong black box attack. However, the evaluation is subject to concern, as the performance barely dips below the benign performance after attack. A simple test to check any problems with the evaluation would be to intentionally inject adversarial examples and check if the performance is still retained [2].

For APGD-ce , APGD -dlr and square, the generated samples for each iteration always reach the specified radius of the ℓp ball. Fab-t tries to find minimally distorted adversarial examples, so it is not quite relevant here.

For APGD-ce and APGD-dlr, it never gives up searching for a better sample even though it already find an effective one in the early iterations. The final returned one will always be the adversarial ones that have defeated the net most recently. In other words, suppose the method finds an effective one in iteration 1. It will continue to search for a better sample to maximize the loss until it reaches 100 iterations. Of course, it might be possible that one with a bigger loss turns out not to be effective. The default parameter to deal with this is best_loss=False in autopgd_base.py. If it finds another effective one at iteration 80 and afterward there is no effective one anymore, it will return this sample at iteration 80.

For square, on the contrary, it will not search for a better sample once it already found an effective one in the early iterations. Unfortunately, the example is very unstable and if fed once again, it will be classified correctly.  In other words, the attacker is fooled by the defender.
This might raise some concerns.

We make the following arguments regarding this. The square attack might have problems on its own and need further improvement. However, there is nothing wrong with our evaluation as it is black and simple to use, and the comparison is fair. In fact, although we are in the adversarial defense community and are striving to propose an effective method to defend, the attacker side can also benefit from our progress in that we help them identify potential pitfalls in the attack. That is another contribution of our paper.

---

### Meta-Review · Area_Chair_6YWz · 2023-12-06

**Metareview:**

The authors suggest a simple defense strategy against adversarial examples using randomization and feature transformation. They seemingly achieve SOTA results against Auto-Attack. Given the long history of wrong results in the adversarial defense literature, it is the responsibility of the authors to make sure that the evaluation is correct and provide all kind of santiy checks in particular if the results are significant improvements compared to SOTA. All reviewers are concerned about the validity of the results and suggest rejection. Unfortunately, the authors did not provide evidence in the rebuttal that their evaluation is correct and thus this is a clear reject. Auto-Attack is known to be reliable for evaluation of deterministic defenses but it can have problems with stochastic ones.

The reviewers have pointed out simple sanity checks which the authors did not do. The most simple one is to check at which radius the robust accuracy goes to zero.

Another one is to make the network deterministic and then do a transfer attack using the adversarial samples from the deterministic model for the stochastic one.

I encourage the authors to do all this checks following existing guidelines and if their results survive these tests to resubmit together with code and models so that independent parties can check the results before publication.

**Justification For Why Not Higher Score:**

There are strong concerns that the evaluation of adversarial robutness is not valid.

**Justification For Why Not Lower Score:**

N/A

---

### Decision · Program_Chairs · 2024-01-16

Reject